# Methicillin-resistant *Staphylococcus aureus* along the beef production line: Phenotypic resistance and *mecA* phylogeny in two ethiopian municipal abattoirs

**Samuel Abie**[1◉], **Solomon Lulie Abey**[2◉*], **Gashaw Getaneh Dagnaw**[2‡], **Wassie Molla**[2‡], **Mebrie Zemene Kinde**[2‡], **Seleshe Nigatu**[2‡], **Mebrat Ejo**[3], **Habtamu Tassew**[1], **Anmaw Shite Abat**[2], **Eyerusalem Belay**[2], **Shimelis Dagnachew**[2], **Abebe Tesfaye Gessese**[2], **Takele Adugna**[2], **Bemrew Admassu Mengistu**[2], **Yitayew Demessie**[2], **Melkie Dagnaw Fenta**[2], **Asnakew Mulaw Berihun**[2], **Wudu Temesgen Jemberu**[2,4], **Adugna Berju**[2], **Getaw Deresse Tadesse**[5], **Yoseph Kerie Kebede**[6], **Kenaw Birhanu**[5], **Takele Abayneh**[5], **Esayas Gelaye**[7◉], **Abebe Belete Bitew**[2◉]

1 School of Veterinary Medicine, Bahir Dar University, Bahir Dar, Ethiopia, 2 College of Veterinary Medicine and Animal Sciences, University of Gondar, Gondar, Ethiopia, 3 Vaccine and Diagnostics Research & Development Division, Armauer Hansen Research Institute, Addis Ababa, Ethiopia, 4 International Livestock research institute (ILRI), Addis Ababa, Ethiopia, 5 National Veterinary Institute, Bishoftu, Ethiopia, 6 Bahir Dar Animal Health Investigation and Diagnostic Laboratory, Bahir Dar, Ethiopia, 7 Food and Agriculture Organization (FAO) of the United Nations, Sub-Regional Office for Eastern Africa, Addis Ababa, Ethiopia

◉ These authors contributed equally to this work.
‡ These authors also contributed equally to this work.
* Solomonlulie@gmail.com

## Abstract

*Staphylococcus aureus (S. aureus)* is a common zoonotic bacteria, responsible for a wide range of infections and is well known for developing resistance to multiple antibiotics. In Ethiopia, information on methicillin-resistant *S. aureus* (MRSA), particularly from a One-Health perspective, is limited. This study aimed to detect *S. aureus*, identify MRSA strains, and assess their antibiogram patterns in swab samples collected from two municipal abattoirs in Northwest Ethiopia. A cross-sectional study was conducted between January 2021 and April 2022. A total of 150 swab samples were collected from beef carcasses, abattoir equipment, surfaces, workers' hands and clothes. Isolation and identification of *S. aureus* followed ISO 6888−2 standards. Antimicrobial susceptibility was tested against ten commonly used antibiotics using the disk diffusion method. Conventional PCR was used for detection of the *mecA* gene and Sanger method was used for sequencing. Overall, *S. aureus* was isolated from 25.3% (38/150) of the samples. The prevalence of *S. aureus* was 27.1% in beef carcasses, 26.9% in abattoir workers, and 23.1% in the abattoir environment. The prevalence was 22.7% in Bahir Dar and 28.0% in Debre Markos abattoirs. The highest detection rate (35.7%) was from workers' hands and hooks, while the lowest

**Data availability statement:** All relevant data are within the manuscript and its Supporting Information files.

**Funding:** The study was financed by the mega research project from the University of Gondar research budget code 6223/2020, funded by the University of Gondar. There was no additional external funding received for this study. The funders had no role in study design, data collection and analysis, decision to publish, or preparation of the manuscript.

**Competing interests:** The authors have declared that no competing interests exist.

(11.1%) was from splitting axes. All isolates were susceptible to gentamicin but resistant to penicillin and methicillin. Multidrug resistance was observed in 60.5% of the isolates. Sequencing and phylogenetic analysis of the *mecA* gene showed that the current isolates were highly similar and clustered closely with *mecA* from *Staphylococcus capitis* and *Staphylococcus fleurettii*, but were distinct from other *S. aureus* strains. The detection of *S. aureus* and MRSA in beef carcasses, abattoir environments, and workers highlights potential risks to workers, consumers, and surrounding environments exposed to abattoir waste. Strengthening hygiene and sanitary practices in abattoirs is essential within a One Health framework.

## Introduction

Foodborne diseases (FBDs) are caused by infectious or toxic agents in contaminated food or water [1]. Of more than 250 foodborne diseases, about two-thirds are attributed to pathogenic bacteria [2]. Common bacterial pathogens include Staphylococcus aureus (S. aureus), Escherichia coli, Listeria monocytogenes, Salmonella, and Campylobacter species, all of which can lead to severe illness and even death [3]. The World Health Organization [4] estimates that up to 30% of people in developed countries suffer from foodborne diseases annually, while in developing countries, about 2 million deaths per year are linked to these infections.

S. aureus is among the most common bacteria implicated in foodborne illnesses, particularly in foods of animal origin such as meat and milk [5–7]. *S. aureus* can produce heat-stable enterotoxins in food before it is eaten, and these toxins can cause food poisoning and gastrointestinal illness. [1,8,9]. In animals, S. aureus is also a leading cause of mastitis [10].

Taxonomically, S. aureus is a Gram-positive, non-spore-forming, catalase- and coagulase-positive bacterium in the family Staphylococcaceae [11]. It is a natural commensal of human and animal skin, hair, nostrils, and mucous membranes [12]. It can cause opportunistic infections following breaches in skin or mucosal barriers [13]. In abattoir settings, S. aureus may be present on animals, equipment, workers, and facility surfaces, contributing to contamination and cross-contamination [14]. Human infection can occur through consumption of contaminated animal products such as meat, milk, and eggs [1,15].

Isolation and identification of S. aureus can be achieved through phenotypic methods such as Gram staining, hemolysis on blood agar, biochemical tests, and molecular techniques [16,17]. Several Ethiopian studies have reported variable prevalence rates of S. aureus in abattoirs, ranging from 9.4% to over 70% depending on the source, hygiene conditions, and sampling methods [6,7,18–20]. Such variation reflects differences in contamination levels and slaughtering practices.

A major public health concern with S. aureus is its ability to develop antimicrobial resistance. Methicillin-resistant S. aureus (MRSA), carrying genes such as *mecA*, has become a global problem, complicating infection control and treatment [21,22]. Previous studies in Ethiopia and elsewhere have reported methicillin resistance rates

of up to 100% in S. aureus isolates [23,24]. MRSA has been detected in meat from livestock species including pork, beef, and chicken [25,26].

These findings emphasize the public health importance of MRSA and the need for strict food hygiene practices to limit its spread. However, in Ethiopia, particularly in the Amhara Regional State, data on S. aureus and MRSA from abattoir settings remain scarce. This study was therefore designed to detect S. aureus, identify methicillin-resistant strains, and assess their antimicrobial resistance profiles in two municipal abattoirs, using a One Health perspective.

## Materials and methods

### Description of the study area

The study was conducted at the municipal abattoirs of Bahir Dar and Debre Markos cities, Amhara Region, Northwest Ethiopia. Bahir Dar city is located between 11°25′19″–11°57′07″ N and 37°14′35″–37°29′07″ E, at an altitude of 1,500–2,600 meters above sea level (masl). The city receives 1,200–1,600 mm annual rainfall and has a temperature range between 15–33°C [27].

Debre Markos town, the capital of East Gojjam Zone, is located about 300 km northwest of Addis Ababa and 265 km southeast of Bahir Dar. It lies between 10°17′00″–10°21′30″ N and 37°42′00″–37°45′30″ E, at 2,350–2,500 masl. The city receives an average of 1,380 mm rainfall annually, with a temperature ranging from 15°C to 22°C [28].

Cattle slaughtered in the two abattoirs originated from nearby districts. During the study period, an average of 13 cattle were slaughtered daily at Bahir Dar and 8 at Debre Markos abattoirs. The workforce consisted of four veterinarians and 13 abattoir workers at Bahir Dar, and one veterinarian and 10 workers at Debre Markos abattoirs. Veterinarians conducted ante- and post-mortem inspections, while workers performed other tasks. Neither abattoir had a structured slaughter process with distinct stages such as stunning, bleeding, skinning, evisceration, or carcass cutting.

### Study population and design

A cross-sectional design was used from January 2021 to November 2021. The study population included for this study were beef carcasses, abattoir equipment (knives, splitting axes, cutting tables, and hooks), slaughterhouse walls, and abattoir workers (hands and clothes).

### Sampling and sample size considerations

Two municipal abattoirs (Bahir Dar and Debre Markos) were purposively selected based on functionality, accessibility and proximity to microbiology laboratories to allow timely sample processing. Within each abattoir we used a purposive, systematic swabbing approach to capture potential contamination points along the beef line (carcass sites, knives, axes, hooks, cutting tables, walls) and personnel (hands and clothes). The aim was to obtain representative swabs across these distinct sampling units rather than to estimate prevalence for a single homogeneous population. A sample size of 150 swabs was chosen to estimate the prevalence of S. aureus with acceptable precision given logistical constraints (field access, laboratory capacity, and budget). We allocated swabs proportionally across sample types to capture contamination at multiple points of the beef line: beef carcasses (59), knives (15), cutting tables (13), hooks (14), walls (14), splitting axes (9), workers' hands (14), and workers' clothes (12).

### Sample collection and transportation

Swabs were collected according to ISO 6888−2 [16]. Swab samples from beef carcass were taken from the abdomen (flank), thorax (lateral), crutch (tail and anal area), and breast (lateral) using a sterile 10 × 10 cm template. A sterile cotton-tipped swab (2 × 3 cm) moistened with buffered peptone water (Oxoid Ltd., England) was rubbed horizontally and vertically over the surface.

Environmental samples were collected from knives, splitting axes, hooks, cutting tables, and walls, while worker samples were taken from hands and clothes during slaughter operations. After swabbing, shafts were broken off, leaving cotton swabs in test tubes. All samples were transported under aseptic conditions in an ice box to Bahir Dar Animal Health Investigation and Diagnostic Laboratory or the College of Veterinary Medicine and Animal Sciences (CVMAS), University of Gondar.

## Isolation and identification of *Staphylococcus aureus*

Samples were inoculated onto blood agar supplemented with 5% sheep blood and incubated aerobically at 37°C for 24 h. Colonies showing hemolysis were subcultured on nutrient agar (HiMedia Lab Pvt Ltd, India) for purity. Suspected colonies were further inoculated on mannitol salt agar (HiMedia Lab Pvt Ltd, India) and incubated at 37°C for 24 h [29]. The cultures that produced pink, yellow or white colonies on mannitol salt agar were considered as *Staphylococcus* species.

Identification was carried out using colony morphology, hemolysis patterns, gram staining, mannitol fermentation, catalase, and coagulase tests. Isolates of *Staphylococcus* species that fermented mannitol, showed catalase activity, and coagulated rabbit plasma were considered as *S. aureus*. Confirmed isolates were stored in nutrient broth (HiMedia Lab Pvt Ltd, India) with 20% glycerol at −20°C until further analysis.

## Antibiogram assessment of *Staphylococcus aureus*

The antimicrobial susceptibility of *S. aureus* isolates was assessed against ten commonly used antimicrobials in the study areas using the disk diffusion method following CLSI guidelines (CLSI, 2018; CLSI, 2021). The antibiotics tested were grouped into their classes as β-lactam antibiotics included ampicillin (AMP, 10 µg), methicillin (MET, 5 µg), penicillin G (P, 10 IU), and ceftazidime (CAZ, 30 µg). The tetracycline class included doxycycline (DO, 30 µg). The macrolide class included erythromycin (E, 15 µg). The aminoglycoside class included gentamicin (CN, 10 µg). The lincosamide class included clindamycin (CD, 2 µg). The sulfonamide class included co-trimoxazole (COT, 25 µg). The quinolone class included norfloxacin (NX, 10 µg).

Bacterial suspensions were adjusted to 0.5 McFarland standard and inoculated onto Mueller–Hinton agar plates (HiMedia Lab Pvt Ltd, India). Antibiotic disks were placed on the surface, and plates were incubated at 37°C for 24 h. Inhibition zones were measured in millimeters, and isolates were classified as resistant, intermediate, or susceptible [30,31].

## Molecular detection of methicillin-resistant *Staphylococcus aureus*

Fifteen *S. aureus* isolates were selected for *mecA* gene testing. These isolates were taken from all sample types (carcasses, workers, equipment, and environment) and from both abattoirs to ensure good representation of the study population. The *mecA* gene, which encodes penicillin-binding protein 2a (PBP2a), was analysed at the National Veterinary Institute in Bishoftu, Ethiopia. Only 15 isolates were tested because the laboratory and project budget could not support molecular testing of all 38 isolates.

Genomic DNA was extracted from *S. aureus* using DNeasy Blood and Tissue Kits (QIAGEN, Germany). *S. aureus* isolates were transferred to 1.5 ml Eppendorf tubes. Bacteria were lysed with AL buffer, and the suspension was carefully transferred to a DNeasy Mini spin column in 2 ml collection tubes. Following the addition of AW1 and AW2 washing buffers, AE buffer was added to a new labeled Eppendorf tube. DNA was eluted and stored at 4°C until PCR amplification.

The extracted genomic DNA of *S. aureus* isolates were amplified by conventional PCR for the detection of *mecA* gene using the forward primer *mecA*-F (5'-AAAATCGATGGTAAAGGTTGGC-3') and the reverse primer *mecA*-R (5'-AGTTCTGCAGTACCGGATTTGC-3') [32]. Amplification of DNA was carried out in a total volume of 22 µl PCR reaction mixtures containing 3µl of nuclease-free water, 10µl of IQ super mix, 2µl of each primer and 5µl of template DNA. The amplification was performed using a thermal cycler (2720, Applied BioSystems, USA) with a PCR protocol of an initial

denaturation at 94°C for 5 min, followed by 35 cycles of denaturation at 94°C for 30 sec, annealing at 54°C for 30 sec, and extension at 72°C for 30 sec, and with a final extension at 72°C for 7 minutes.

PCR products were separated using 2% agarose gel in 1X TAE buffer at 120 V for 60 min. Gels were stained with GelRed (Biotium, inc.) and visualized under UV light. Four µl of loading dye was mixed with PCR products, and 10µl of this mixture was poured into each well. A 100 bp DNA ladder was used as a marker.

A known *mecA*-positive *S. aureus* strain was used as the positive control in each PCR run, and nuclease-free water served as the negative control. An extraction control was also included to check for contamination during DNA preparation. All PCR reactions were repeated to confirm the consistency of the results.

### Sequencing and phylogenic analysis

The *mecA* gene of the tested isolates was sequenced at LGC Genomics (Germany). The amplified *mecA* gene fragments were purified with the Wizard® SV Gel and PCR Clean-Up System (Promega, Germany). The purified PCR products of the *mecA* gene from S. aureus isolates were subsequently sequenced by the Sanger method.

The evolutionary history of the *mecA* gene of the identified strains was performed via comparative phylogenetic analysis. The *mecA* gene sequences of the strains were aligned with those of *S. aureus* strains gene sequences obtained from GenBank using NCBI BLASTn. Multiple sequence alignments were performed using MUSCLE program, and the evolutionary history was inferred using the Neighbor-Joining method [33]. For phylogeny test, bootstrap method with 1000 replicates was used [34]. The evolutionary distances were computed using the Maximum Composite Likelihood method [35] and are in the units of the number of base substitutions per site. The phylogenic tree was constructed using MEGA11 [35].

### Ethical consideration

The ethical clearance was obtained from the College of Veterinary Medicine and Animal Sciences research ethics review committee, University of Gondar (Ref. CVMAS.sc-07/2020). Permission was obtained from Bahir Dar and Debre Markos municipal authorities. Informed verbal consent was obtained from abattoir workers prior to the collection of hand and cloth swab samples. The consent process was explained in the local language (Amharic), ensuring that participants fully understood the purpose and procedure of the study. Verbal consent was documented by the researcher through written records confirming each participant's agreement, and samples were collected only from those who voluntarily consented to participate. The recruitment and sampling period was conducted from January 1, 2021, to April 30, 2021.

### Data management and analysis

Data were entered and cleaned using Microsoft excel spreadsheet. Descriptive statistics were performed using statistical tools in Stata version 17. Bioinformatics tools were used to sequence and analyze the purified PCR products of the *S. aureus* isolate *mecA* gene. To understand the evolutionary relationships among the *mecA* gene of the various *Staphylococcus* species and strains, phylogenetic analysis was computed using MEGA 11.

## Results

### Isolation and identification of *Staphylococcus aureus*

Out of 150 swab samples, 38 (25.3%) were positive for S. aureus. The proportion of isolation at Debre Markos abattoir was 28.0% (21/75), compared to 22.7% (17/75) in Bahir Dar abattoir. By sample source, S. aureus was isolated from 27.1% (16/59) of carcass samples, 23.1% (15/65) of environmental samples, and 26.9% (7/26) of abattoir worker samples (Table 1).

**Table 1. Proportion of *S. aureus* isolates from Bahir Dar and Debre Markos municipal abattoirs, January 2021 to April 2022.**

| Sample source | Sample location | Number examined | Number positive (%) | Total (%) |
|---|---|---|---|---|
| **Beef carcass** | Bahir Dar | 29 | 7 (24.1) | *16 (27.1)* |
| | Debre Markos | 30 | 9 (30) | |
| **Abattoir environment** | Bahir Dar | 33 | 8 (24.2) | *15 (23.1)* |
| | Debre Markos | 32 | 7 (21.9) | |
| **Abattoir worker** | Bahir Dar | 13 | 2 (15.4) | *7 (26.9)* |
| | Debre Markos | 13 | 5 (38.5) | |
| **Sub-total** | Bahir Dar | 75 | 17 (22.7) | 38 (25.3) |
| | Debre Markos | 75 | 21 (28.0) | |
| **Total** | | **150** | **38 (25.3)** | **38 (25.3)** |

## Distribution of *S. aureus* isolates across sample sources

The proportion of *S. aureus* isolation varied across sample sources, ranging from 11.1% to 35.7% (Table 2). The highest rates were observed on abattoir workers' hands (35.7%) and hooks (35.7%), while the lowest was from splitting axes (11.1%). Notably, S. aureus was detected from every type of sampling source, including carcasses, equipment, surfaces, and workers' hands and clothes.

From 59 swab samples that were collected from beef carcass, 16(27.1%) were tested positive for *S. aureus*. Of which, 26.7% (4/15), 28.6% (4/14), 31.3% (5/16) and 21.4% (3/14) were isolated from abdomen, thorax, crutch and breast regions of the beef carcass, respectively (Table 2).

## Antibiotic susceptibility profiling

Antibiotic susceptibility testing revealed different resistance patterns among the 38 isolates (Fig 1). All isolates were susceptible to gentamicin (100%), while all were resistant to penicillin and methicillin (100%). High susceptibility was also observed to norfloxacin (86.8%) and co-trimoxazole (63.2%). In contrast, resistance was frequent to doxycycline (63.2%), clindamycin (60.5%), and ampicillin (57.9%).

Multidrug resistance was detected from 60.5% (23/38) of the isolates. Among these, 16 isolates were resistant to three classes of antibiotics, five isolates to four classes, one isolate to five and another one to six classes of antibiotics (Table 3).

**Table 2. Distribution of *S. aureus* isolates across sample sources.**

| Sample source | Category | Number examined | Number positive (%) |
|---|---|---|---|
| Carcass swab | Abdomen | 15 | 4 (26.7) |
| | Thorax | 14 | 4 (28.6) |
| | Crutch | 16 | 5 (31.3) |
| | Breast | 14 | 3 (21.4) |
| Workers | Hands | 14 | 5 (35.7) |
| | Clothes | 12 | 2 (16.7) |
| Environment/equipment | Knife | 15 | 4 (26.7) |
| | Splitting axe | 9 | 1 (11.1) |
| | Cutting table | 13 | 3 (23.1) |
| | Hooks | 14 | 5 (35.7) |
| | Walls of slaughter area | 14 | 2 (14.3) |

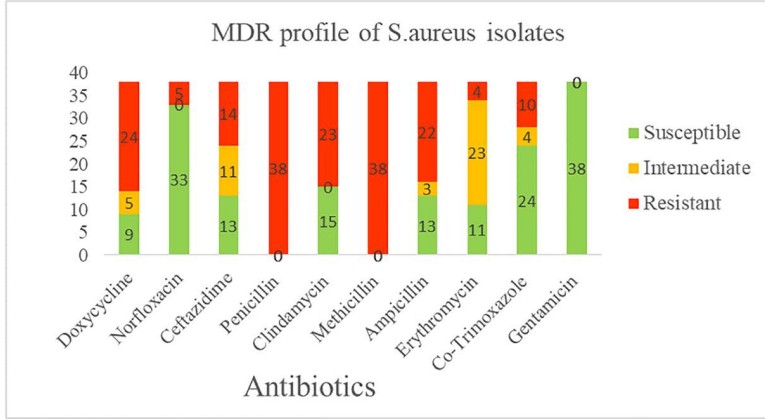

**Fig 1. Antibiotic susceptibility profiling of *Staphylococcus aureus* isolates (n = 38).**

## Molecular detection of methicillin resistant *Staphylococcus aureus*

The PCR screening of 15 isolates (representatives from each sample types) revealed that 5 (33.3%) carried the mecA gene (Table 4, Fig 2). Of these, three were from carcasses and two from workers. No *mecA* gene was detected in environmental isolates. Detection rates were higher in Debre Markos (60%) than Bahir Dar (20%).

## Sequencing and phylogenetic analysis of *mecA*

Sequence analysis of the two *S. aureus* isolates yielded a 508 bp partial coding sequence, confirming the presence of the *mecA* gene. These sequences were submitted to GenBank and assigned the accession numbers PP735247 and PP735248.

**Table 3. Multidrug resistance profiles of S. aureus isolates.**

| No of Antibiotics class | Resistance pattern | Number of resistance isolates sub-total (%) | Total Number of resistance isolates in antibiotic class combination (%) |
|---|---|---|---|
| Three | β-lactam, Tetracycline, Lincosamides | 10 (26.32) | 16 (42.11) |
| | β-lactam, Tetracycline, Macrolides | 1 (2.63) | |
| | β-lactam, Tetracycline, Sulfonamide | 1 (2.63) | |
| | β-lactam, Lincosamides, Sulfonamide | 3 (7.9) | |
| | β-lactam, Lincosamides, Quinolones | 1 (2.63) | |
| Four | β-lactam, Tetracycline, Lincosamides, Co-Trimoxazole | 3 (7.9) | 5 (13.16) |
| | β-lactam, Tetracycline, Lincosamides, Quinolones | 1 (2.63) | |
| | β-lactam, Tetracycline, Quinolones, Macrolides | 1 (2.63) | |
| Five | β-lactam, Tetracycline, Lincosamides, Quinolones, Sulfonamide | 1 (2.63) | 1 (2.63) |
| Six | β-lactam, Tetracycline, Lincosamides, Quinolones, Sulfonamide, Macrolides | 1 (2.63) | 1 (2.63) |
| Total | | | 23(60.53) |

**Table 4. Detection of *mecA* gene among S. aureus isolates.**

| Sample source | Sample area | No. tested | *mecA* positive (%) | Total *mecA* positive (%) |
|---|---|---|---|---|
| **Beef carcass** | Bahir Dar | 4 | 1 (25) | ***3 (42.9)*** |
| | Debre Markos | 3 | 2 (66.7) | |
| **Abattoir environment** | Bahir Dar | 4 | 0 (0.0) | ***0 (0)*** |
| | Debre Markos | 1 | 0 (0.0) | |
| **Abattoir worker** | Bahir Dar | 2 | 1 (50.0) | 2 (66.7) |
| | Debre Markos | 1 | 1 (100.0) | |
| **Sub-total** | Bahir Dar | 10 | 2 (20.0) | 5 (33.3) |
| | Debre Markos | 5 | 3 (60.0) | |
| **Total** | | **15** | **5 (33.3)** | **5 (33.3)** |

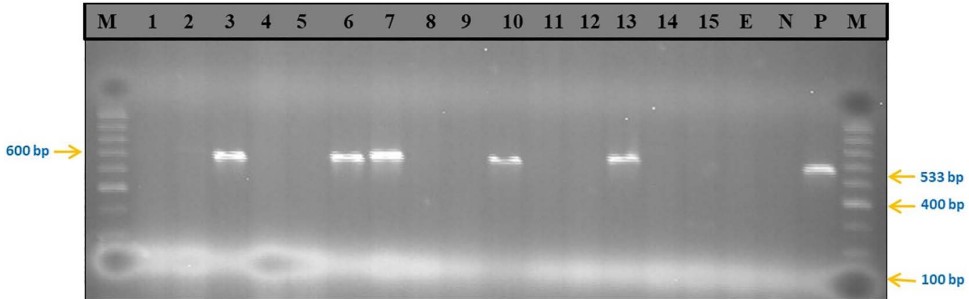

**Fig 2. Agarose gel electrophoresis of *mecA* gene (533 bp).** M = DNA marker (100 bp); Lanes 1–15 = test samples; E = extraction control; N = negative control, and P = positive control.

The phylogenetic analysis revealed the evolutionary relationships among the *mecA* gene of the various *Staphylococcus* species and strains as shown in Fig 3. The close clustering of *mecA* of the strains found in this study demonstrated that they have a high level of genetic similarity. The *mecA* genes of the strains identified in this study were distantly related to the *mecA* genes of other *Staphylococcus aureus* strains, such as MK341125.1, MK659556, MH798869, and MW052031. In contrast, they clustered closely with *mecA* genes of other *Staphylococcus* species such as *Staphylococcus capitis* (NG 047941.1) and *Staphylococcus fleurettii* (HE978796.1).

## Discussion

The overall prevalence of S. aureus isolation in this study was 25.3%, with a higher rate in Debre Markos (28.0%) than in Bahir Dar (22.7%). This prevalence is lower than previous reports of 36–49% in Asella, Mekelle, and Bishoftu municipal [20,36,37], but higher than findings of 9–13% in Addis Ababa [23,38]. Such observed variation may be due to differences in hygiene practices, water availability, sterilization procedures, difference in sampling strategy, and variations in the stages of the slaughtering process at which samples were collected.

The proportion of S. aureus in beef carcasses (27.1%) was more or less in line with the previous Ethiopian findings (34.3%; [19]) and to South Africa (20.4%; [39]). However, it was much lower than the finding of Sadiq *et al.* [11] who reported as 75% in Pakistan and Rahimi *et al.* [40] who reported 57.5% in Iran, while higher than 5–12% reported in Bishoftu and Addis Ababa [38,41]. These differences may reflect variations in sample size, hygiene of the slaughtering process, and geographic practices.

Environmental contamination was also evident in the current findings. Knives (26.7%) were positive for *S. aureus*. This finding is relatively lower than the 33.3% *S. aureus* prevalence from knife swabs in Addis Ababa [38] and 50% in Asella

**Fig 3. Phylogenetic tree of *mecA* gene sequences of isolates from this study compared with reference strains.** Bootstrap values (>50%) from 1,000 replicates are shown. Strains from this study are marked with red triangles. The evolutionary history was inferred using the Neighbor-Joining method and the optimal tree is presented. The percentage of replicate trees in which the associated taxa clustered together in the bootstrap test (1000 replicates) are shown next to the branches. The evolutionary distances were computed using the Maximum Composite Likelihood method and are in the units of the number of base substitutions per site. This analysis involved 14 nucleotide sequences. Codon positions included were 1st+2nd+3rd+Non-coding. All ambiguous positions were removed for each sequence pair (pairwise deletion option). There were a total of 534 positions in the final dataset.

[19] while higher than the 22.5% prevalence in Pakistan [11]. Hooks (35.7%) were also particularly contaminated with *S. aureus*, consistent with the 33.3% finding from Addis Ababa, Ethiopia [38]. The current findings was higher than the 15% reported by Adugna *et al.* [23] from Addis Ababa while lower than the 60% result that reported by Hassan *et al.* [19] from Assela, Ethiopia. Splitting axes (11.1%) and abattoir walls (14.3%) showed the lowest contamination, though this still indicates cross-contamination potential. Such contamination is linked to inadequate cleaning, lack of sterilization, and repeated contact with carcasses and workers' hands.

Notably, workers' hands (35.7%) and clothes (16.7%) were also positive for S. aureus. Similar patterns have been reported in Asella [18,36]. Higher prevalence in hands compared to clothes likely reflects direct contact with carcasses and contaminated equipment; however, it may also be partly due to the natural colonization of *S. aureus* on human skin. These findings emphasize the role of personal hygiene and proper use of protective equipment in preventing carcass contamination.

In this study, comparison of *S. aureus* isolation rates from knife, table, axe, hook, abattoir workers' hand and cloth, wall of abattoir houses and beef swabs in abattoirs revealed that the highest proportion of isolation was recorded in workers' hand and hook swabs. This might be due to meat contamination occurring at various stages of the slaughtering process, particularly through handling hooks with contaminated hands. Next to the hook and workers' hand swabs, high number of *S. aureus* was isolated from beef carcass swab than any other swab samples. This could be attributed to the highest sample size of the beef carcass samples than other sources of samples and the contaminated personnel at abattoir might have also contributed to beef carcass contamination.

All isolates were susceptible to gentamicin, consistent with reports from Bishoftu and Addis Ababa [38,41]. However, the current report is disagree with previous studies from Mekelle and Addis Ababa that found partial resistance [20,42]. This suggests gentamicin remains effective in the study areas, possibly due to its limited use. High susceptibility was also observed to norfloxacin (86.8%), similar to Debre- Zeit findings [43].

In contrast, all isolates were resistant to penicillin and methicillin. This mirrors findings from Ethiopia [23,37] and other countries [44], but is higher than resistance levels reported in the USA (74%; [45]) and Ethiopia (52%; [46]). Such widespread resistance likely reflects long-term, indiscriminate use of β-lactams in both veterinary and human medicine. Resistance to doxycycline (63.2%), clindamycin (60.5%), and ampicillin (57.9%) further highlights growing antimicrobial resistance pressure.

In this study, more than 60% of the isolates exhibited multidrug resistance (MDR), in line with reports from Ethiopia and Europe [18,44,47]. This trend indicates that *S. aureus* strains circulating in abattoirs are adapting to multiple antibiotics, including commonly used β-lactams in both human and veterinary medicine. Such adaptation increases the likelihood of treatment failures in humans and animals and underscores the need for prudent antimicrobial use and strengthened stewardship across sectors.

PCR confirmed the presence of the *mecA* gene in 33.3% of tested isolates. Detection was higher in Debre Markos (60%) than in Bahir Dar (20%), suggesting geographic variability in the distribution of MRSA strains. Similar detection rates have been reported in pig and chicken carcasses (22.6%; [24]), while higher prevalence was found in beef in Pakistan (63%; [11]).

Interestingly, although all isolates were phenotypically resistant to methicillin, only some carried the *mecA* gene. This discrepancy could be explained by the presence of alternative resistance determinants such as mecC, mecB, or mecD [48–50]. Similar findings of methicillin resistance without *mecA* have been reported in Europe and North America [47,51]. This highlights the need for expanded molecular screening beyond *mecA*.

Findings of the phylogenetic analysis provide important information on the evolutionary relationships among the *mecA* gene of *Staphylococcus* species and strains. The observed close relationship of the *mecA* gene of strains identified in this study indicates a high degree of genetic similarity between MRSA-BahirDar-01–2021 (PP735247) and MRSA-BahirDar-02–2021 (PP735248), suggesting a common evolutionary lineage and the clonal expansion of a specific strain within the study area. In contrast, the *mecA* sequences of the strains found in this investigation were distantly related to other global *S. aureus* strains, such as MK341125, MK659556, MH798869, and MW052031. This indicates that these strains carry genetic variations in their *mecA* sequences compared to international clones, which is consistent with the findings of Hanssen *et al*. [52], who found variation of the *mecA* gene by location alongside sequence conservation observed among isolates from the same geographic region. Such divergence may reflect local epidemiological trends and regional adaptations to antibiotic pressure, aligning with findings that highlight the role of environmental factors in shaping resistance profiles [53]. Finally, the close relationship between the *mecA* of strains identified in this study and the *mecA* of other *Staphylococcus* species (specifically *S. fleurettii*) suggested that the resistance gene likely originated from these commensal species through horizontal gene transfer, as observed in the study of Schwendener and Perreten [54].

The limitations of this study were the small sample size, and the restriction of molecular analysis to the *mecA* gene without testing for other methicillin resistance determinants. These constraints may have led to an underestimation of the real prevalence and genetic diversity of MRSA in the study area.

## Conclusion

The detection of *Staphylococcus aureus* and *methicillin-resistant S. aureus* (MRSA) in beef carcasses, abattoir environments, and abattoir workers highlights a substantial public health concern. Slaughterhouse contamination represents a critical point for the dissemination of resistant bacteria, posing risks to workers, consumers, and surrounding communities through direct contact and environmental exposure. The high prevalence of antimicrobial resistance observed in this study underscores the need for strengthened abattoir hygiene and sanitation practices, including regular cleaning and disinfection of equipment and surfaces, systematic use of personal protective equipment, and targeted training of abattoir personnel. In addition, continuous antimicrobial resistance surveillance and the promotion of judicious antimicrobial use in both veterinary and human medicine are essential. Implementing these measures within a One Health framework will be crucial for mitigating MRSA transmission along the food chain and safeguarding animal, human, and environmental health.

## Supporting information

**S1 File. S1 file Antibiogram profiles of *S. aureus* isolates.**
(PDF)

**S1 Fig. S1 raw image.**
(PDF)

## Acknowledgments

The authors would like to acknowledge the University of Gondar, College of Veterinary Medicine and Animal Science, the Bahir Dar Animal Health Investigation and Diagnostic Laboratory, and the National Veterinary Institute. The sequences published in this article have been generated through the Sanger Sequencing Services of the Animal Production and Health Sub-programme of the Joint FAO/IAEA Centre in Vienna, Austria. We also extend our gratitude to the laboratory technicians in each institution for their assistance.

## Author contributions

**Conceptualization:** Solomon Lulie Abey, Gashaw Getaneh Dagnaw, Wassie Molla, Mebrie Zemene Kinde, Seleshe Nigatu, Shimelis Dagnachew, Takele Adugna, Bemrew Admassu Mengistu, Wudu Temesgen Jemberu, Adugna Berju, Abebe Belete Bitew.

**Data curation:** Samuel Abie, Solomon Lulie Abey, Gashaw Getaneh Dagnaw, Wassie Molla, Mebrie Zemene Kinde, Seleshe Nigatu, Mebrat Ejo, Habtamu Tassew, Shimelis Dagnachew, Abebe Tesfaye Gessese, Takele Adugna, Bemrew Admassu Mengistu, Yitayew Demessie, Melkie Dagnaw Fenta, Asnakew Mulaw Berihun, Wudu Temesgen Jemberu, Adugna Berju, Esayas Gelaye, Abebe Belete Bitew.

**Formal analysis:** Samuel Abie, Solomon Lulie Abey, Gashaw Getaneh Dagnaw, Wassie Molla, Mebrie Zemene Kinde, Seleshe Nigatu, Mebrat Ejo, Habtamu Tassew, Anmaw Shite Abat, Eyerusalem Belay, Abebe Tesfaye Gessese, Takele Adugna, Bemrew Admassu Mengistu, Yitayew Demessie, Melkie Dagnaw Fenta, Asnakew Mulaw Berihun, Wudu Temesgen Jemberu, Esayas Gelaye, Abebe Belete Bitew.

**Funding acquisition:** Solomon Lulie Abey, Gashaw Getaneh Dagnaw, Wassie Molla, Mebrie Zemene Kinde, Seleshe Nigatu, Mebrat Ejo, Esayas Gelaye, Abebe Belete Bitew.

**Investigation:** Samuel Abie, Solomon Lulie Abey, Gashaw Getaneh Dagnaw, Wassie Molla, Mebrie Zemene Kinde, Seleshe Nigatu, Getaw Deresse Tadesse, Yoseph Kerie Kebede, Kenaw Birhanu, Takele Abayneh, Esayas Gelaye, Abebe Belete Bitew.

**Methodology:** Samuel Abie, Solomon Lulie Abey, Gashaw Getaneh Dagnaw, Wassie Molla, Mebrie Zemene Kinde, Seleshe Nigatu, Mebrat Ejo, Habtamu Tassew, Anmaw Shite Abat, Eyerusalem Belay, Shimelis Dagnachew, Abebe Tesfaye Gessese, Takele Adugna, Bemrew Admassu Mengistu, Yitayew Demessie, Melkie Dagnaw Fenta, Asnakew Mulaw Berihun, Wudu Temesgen Jemberu, Adugna Berju, Getaw Deresse Tadesse, Kenaw Birhanu, Takele Abayneh, Esayas Gelaye, Abebe Belete Bitew.

**Project administration:** Solomon Lulie Abey, Abebe Belete Bitew.

**Resources:** Samuel Abie, Solomon Lulie Abey, Gashaw Getaneh Dagnaw, Wassie Molla, Mebrie Zemene Kinde, Seleshe Nigatu, Habtamu Tassew, Bemrew Admassu Mengistu, Wudu Temesgen Jemberu, Kenaw Birhanu, Esayas Gelaye, Abebe Belete Bitew.

**Software:** Samuel Abie, Solomon Lulie Abey, Gashaw Getaneh Dagnaw, Wassie Molla, Mebrie Zemene Kinde, Seleshe Nigatu, Abebe Tesfaye Gessese, Bemrew Admassu Mengistu, Esayas Gelaye, Abebe Belete Bitew.

**Supervision:** Solomon Lulie Abey, Habtamu Tassew, Esayas Gelaye, Abebe Belete Bitew.

**Validation:** Samuel Abie, Solomon Lulie Abey, Gashaw Getaneh Dagnaw, Wassie Molla, Mebrie Zemene Kinde, Seleshe Nigatu, Mebrat Ejo, Habtamu Tassew, Anmaw Shite Abat, Eyerusalem Belay, Shimelis Dagnachew, Abebe Tesfaye

Gessese, Takele Adugna, Bemrew Admassu Mengistu, Yitayew Demessie, Melkie Dagnaw Fenta, Asnakew Mulaw Berihun, Wudu Temesgen Jemberu, Adugna Berju, Getaw Deresse Tadesse, Yoseph Kerie Kebede, Kenaw Birhanu, Takele Abayneh, Esayas Gelaye, Abebe Belete Bitew.

**Visualization:** Samuel Abie, Solomon Lulie Abey, Gashaw Getaneh Dagnaw, Wassie Molla, Mebrie Zemene Kinde, Seleshe Nigatu, Mebrat Ejo, Habtamu Tassew, Anmaw Shite Abat, Shimelis Dagnachew, Abebe Tesfaye Gessese, Takele Adugna, Bemrew Admassu Mengistu, Yitayew Demessie, Melkie Dagnaw Fenta, Asnakew Mulaw Berihun, Wudu Temesgen Jemberu, Adugna Berju, Getaw Deresse Tadesse, Yoseph Kerie Kebede, Kenaw Birhanu, Takele Abayneh, Esayas Gelaye, Abebe Belete Bitew.

**Writing – original draft:** Samuel Abie, Solomon Lulie Abey, Gashaw Getaneh Dagnaw, Wassie Molla, Mebrie Zemene Kinde, Seleshe Nigatu, Abebe Belete Bitew.

**Writing – review & editing:** Samuel Abie, Solomon Lulie Abey, Wassie Molla, Mebrie Zemene Kinde, Seleshe Nigatu, Mebrat Ejo, Habtamu Tassew, Anmaw Shite Abat, Eyerusalem Belay, Shimelis Dagnachew, Abebe Tesfaye Gessese, Takele Adugna, Bemrew Admassu Mengistu, Yitayew Demessie, Melkie Dagnaw Fenta, Asnakew Mulaw Berihun, Wudu Temesgen Jemberu, Adugna Berju, Getaw Deresse Tadesse, Yoseph Kerie Kebede, Kenaw Birhanu, Takele Abayneh, Esayas Gelaye, Abebe Belete Bitew.

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
