## [Decision Letter · Decision Letter 0]

27 Nov 2025

PONE-D-25-53029Identification, Antibiogram assessment and Molecular Detection of Methicillin Resistant Staphylococcus aureus in Beef Line in two Municipal Abattoirs, Northwest EthiopiaPLOS ONE

Dear Dr. Abey,

Thank you for submitting your manuscript to PLOS ONE. After careful consideration, we feel that it has merit but does not fully meet PLOS ONE’s publication criteria as it currently stands. Therefore, we invite you to submit a revised version of the manuscript that addresses the points raised during the review process.

We look forward to receiving your revised manuscript.

Kind regards,

Md. Tanvir Rahman, DVM, MSc (Canada), PhD (UK), FBAS

Academic Editor

PLOS ONE

Journal Requirements:

“The study was financed by the mega research project from the University of Gondar research budget code 6223/2020, funded by the University of Gondar.”

“The study was financed by the mega research project from the University of Gondar research budget code 6223/2020, funded by the University of Gondar.”

5. We note that your Data Availability Statement is currently as follows: [All relevant data are within the manuscript and its Supporting Information files.]

6. Please be informed that funding information should not appear in the Acknowledgments section or other areas of your manuscript. We will only publish funding information present in the Funding Statement section of the online submission form. Please remove any funding-related text from the manuscript.

7. PLOS ONE now requires that authors provide the original uncropped and unadjusted images underlying all blot or gel results reported in a submission’s figures or Supporting Information files. This policy and the journal’s other requirements for blot/gel reporting and figure preparation are described in detail at https://journals.plos.org/plosone/s/figures#loc-blot-and-gel-reporting-requirements and https://journals.plos.org/plosone/s/figures#loc-preparing-figures-from-image-files. When you submit your revised manuscript, please ensure that your figures adhere fully to these guidelines and provide the original underlying images for all blot or gel data reported in your submission. See the following link for instructions on providing the original image data: https://journals.plos.org/plosone/s/figures#loc-original-images-for-blots-and-gels.

8. We note that Figure 1 in your submission contain [map/satellite] images which may be copyrighted. All PLOS content is published under the Creative Commons Attribution License (CC BY 4.0), which means that the manuscript, images, and Supporting Information files will be freely available online, and any third party is permitted to access, download, copy, distribute, and use these materials in any way, even commercially, with proper attribution. For these reasons, we cannot publish previously copyrighted maps or satellite images created using proprietary data, such as Google software (Google Maps, Street View, and Earth). For more information, see our copyright guidelines: http://journals.plos.org/plosone/s/licenses-and-copyright.

a. You may seek permission from the original copyright holder of Figure(s) [#] to publish the content specifically under the CC BY 4.0 license.

Please upload the completed Content Permission Form or other proof of granted permissions as an "Other" file with your submission

Additional Editor Comments (if provided):

Please see the comments of the reviewers amd address them, do experiment where applied.

Reviewers' comments:

Reviewer's Responses to Questions

**Comments to the Author**

1. Is the manuscript technically sound, and do the data support the conclusions?

Reviewer #1: Yes

Reviewer #2: No

2. Has the statistical analysis been performed appropriately and rigorously? 

Reviewer #1: Yes

Reviewer #2: Yes

3. Have the authors made all data underlying the findings in their manuscript fully available?

Reviewer #1: No

Reviewer #2: Yes

4. Is the manuscript presented in an intelligible fashion and written in standard English?

Reviewer #1: Yes

Reviewer #2: Yes

5. Review Comments to the Author

Reviewer #1: Overall Assessment:

The study “Identification, Antibiogram Assessment and Molecular Detection of Methicillin Resistant Staphylococcus aureus in Beef Line in Two Municipal Abattoirs, Northwest Ethiopia“addresses a relevant One Health issue concerning MRSA in abattoirs, which is important for food safety and public health. However, the manuscript has limited novelty and requires improvement in methodological justification, data analysis, and interpretation to meet publication standards. Below are the comments on the article that needs to be addressed before consideration for publication.

Novelty:

The topic is not new; similar studies have been conducted in Ethiopia. The authors should better highlight what new information this study contributes, such as unique molecular or phylogenetic findings.

Methodology:

Sampling strategy and sample size lack statistical justification.

Only 15 isolates were tested for mecA gene—this needs justification.

Phenotypic identification without molecular confirmation (e.g., nuc gene) limits accuracy.

PCR controls and validation details are missing.

Results and Analysis:

Statistical tests (e.g., Chi-square) should be applied to strengthen findings.

The discrepancy between phenotypic and genotypic MRSA detection (100% vs 33.3%) is not well explained.

Discussion:

Too descriptive—should focus more on interpretation and comparison with recent studies.

The phylogenetic results could be better discussed in terms of evolutionary or epidemiological significance.

Conclusion:

Valid but too general. Include clearer recommendations for abattoir hygiene, AMR monitoring, and One Health interventions.

Recommendation:

Major Revision — The manuscript provides useful local data but requires stronger methodological rigor, analytical depth, and clearer discussion of its novel contributions before publication.

Reviewer #2: The authors describe the prevalence of MRSA amongst two Municipal abattoirs in Northwest Ethiopia. While previous studies have described MRSA prevalence amongst abattoirs with variable outcomes, this study evaluates its prevalence in an unexplored region. However, the study is deeply flawed in its overall execution. For instance, it is unclear how these MRSA are identified. Why is methicillin used for screening when it is not the most ideal method to screen for MRSA. It is also not clear why the authors have sequenced the mecA gene, which is quite conserved amongst MRSA strains. I would request the authors to comment on the utility for mecA gene sequence for source attribution or transmission analysis in introduction. In any case the premise for the study and methodology adapted is weak. Perhaps using MLST, spa, SCCmec typing, PFGE or WGS would have been much more appropriate techniques for assessing MRSA transmission patterns amongst workers, environment and the carcasses.

Other comments:

Title: Title is too generalized. Please modify the title to reflect basic outcome of the study

Line 55: kindly rephrase this sentence; It produces preformed enterotoxins in food, which cause food poisoning and gastrointestinal illness. What is meant by pre-formed in this context?

What is the reason for focusing on the mecA gene for sequence analysis?

Line 131-133: Why the authors chose to screen using methicillin rather than cefoxitin which is better recommended for MRSA detection? Methicillin has limited utility to detect MRSA due to heterogeneous expression, high sensitivity to variations in laboratory conditions (inoculum size, incubation time, temperature, and salt concentration in the culture medium), leading to inconsistent or false-negative results. Furthermore, hyperproduction of beta-lactamase can also lead to borderline resistance that is difficult to distinguish with routine methicillin tests. As one of the aims of the study was to screen for mecA gene, this would greatly mis represent the number of MRSA amongst isolates. It would also be useful to include vancomycin in testing.

Line 199: Change Subculture to subcultured

Line 160: Italicize mecA

Remove last two sentences in Data Management and Analysis.

Line 208: Change Antibiogram assessment to antibiotic susceptibility profiling.

Table 3 is very confusing. What is the difference between number of resistant vs total number of resistant isolates? I assume first column describes the combination of resistance against different class combination of antibiotics. Kindly merge last column with first. For example; Three classes of antibiotics (n=16).

Line 219: How were the 15 isolates selected for mecA gene screening? I assume these were on the basis of phenotypic resistance against methicillin/ oxacillin/ cefoxitin?

Lines 229-232: Why the same isolate (MRSA-BahirDar-01-2021) has different accession numbers?

Line 243-250: These are more suited to the methods description and does not mention the results obtained from the analysis.

Line 273-276: The authors discuss how workers hands compared to clothes have a higher S. aureus carriage rate and thus indicate contamination due to carcass handling. Could the higher prevalence also be indicative that it is part of normal flora of the human skin? To infer that this is entirely due to carcass handling may therefore not be accurate.

Line 292-294: Please discuss some data related to long-term, indiscriminate use of β- lactams in both veterinary and human medicine in Ethiopia.

6. PLOS authors have the option to publish the peer review history of their article (what does this mean?). If published, this will include your full peer review and any attached files.

Reviewer #1: **Yes:** Prof. Dr. Tahir Usman

Reviewer #2: No

---

## [Author Response · Author response to Decision Letter 1]

29 Dec 2025

Dear Editor-in-Chief and Reviewers,

We are thankful for the consideration of our manuscript in your journal and for the constructive comments provided, which have significantly improved the quality and clarity of the manuscript. The full, detailed responses to all editorial and reviewer comments are presented in the file name "Response to reviewers", with corresponding line numbers indicated based on the file titled “Revised Manuscript with Track Changes.”

---

## [Decision Letter · Decision Letter 1]

16 Apr 2026

PONE-D-25-53029R1Methicillin-Resistant Staphylococcus aureus Along the Beef Production Line: Phenotypic Resistance and mecA Phylogeny in Two Ethiopian Municipal AbattoirsPLOS One

Dear Dr. Abey,

Thank you for submitting your manuscript to PLOS ONE. After careful consideration, we feel that it has merit but does not fully meet PLOS ONE’s publication criteria as it currently stands. Therefore, we invite you to submit a revised version of the manuscript that addresses the points raised during the review process.

We look forward to receiving your revised manuscript.

Kind regards,

Md. Tanvir Rahman, DVM, MSc (Canada), PhD (UK), FBAS

Academic Editor

PLOS One

Journal Requirements:

Additional Editor Comments :

Please revised the manuscript as suggested.

Comments:

The manuscript addresses an important public health issue by investigating Methicillin-resistant Staphylococcus aureus along the beef production chain. The integration of phenotypic resistance profiling with mecA phylogenetic analysis is valuable and relevant. However, the manuscript requires careful revision to improve clarity, consistency, and scientific accuracy. Several sections contain repetition, inconsistent formatting (particularly for scientific names, gene names, and antibiotic nomenclature), and methodological ambiguities. Addressing these issues will significantly enhance the overall quality and readability of the manuscript.

Lines 166–167: Reference 32 (Mehrotra et al., 2000) does not correspond to the same primers authors used for mecA amplification. The primers described in the manuscript differ from those in the cited reference. Please provide an appropriate and accurate reference.

Additionally, Please provide a clearer, high-resolution PCR gel image and verify the reported product size.

Lines 104–109: This section repeats information presented in the following paragraph. Please remove the redundant text to improve clarity and avoid duplication.

Line 136: The term “Gram” should be written in lowercase (i.e., “gram”).

Lines 144–146: Please group antibiotics according to their respective classes and clearly mention the class names. Additionally, ensure consistency in formatting—antibiotic names should follow a uniform style throughout the manuscript (e.g., consistent capitalization).

Line 152: Scientific names should be written in italics.

Line 160: The sentence is unclear. It is recommended to revise it as: “Genomic DNA was extracted from Staphylococcus aureus.”

Line 183: The DNA extraction method has already been described earlier. Repetition is unnecessary—please remove this section.

Line 250: Scientific names should be written in italics.

Line 251: The gene name mecA is inconsistently formatted (italicized in some places and not in others). Please adopt a consistent style throughout the manuscript.

Reviewers' comments:

Reviewer's Responses to Questions

**Comments to the Author**

1. If the authors have adequately addressed your comments raised in a previous round of review and you feel that this manuscript is now acceptable for publication, you may indicate that here to bypass the “Comments to the Author” section, enter your conflict of interest statement in the “Confidential to Editor” section, and submit your "Accept" recommendation.

Reviewer #1: All comments have been addressed

Reviewer #3: (No Response)

2. Is the manuscript technically sound, and do the data support the conclusions?

Reviewer #1: Yes

Reviewer #3: Yes

3. Has the statistical analysis been performed appropriately and rigorously? 

Reviewer #1: I Don't Know

Reviewer #3: I Don't Know

4. Have the authors made all data underlying the findings in their manuscript fully available?

Reviewer #1: Yes

Reviewer #3: Yes

5. Is the manuscript presented in an intelligible fashion and written in standard English?

Reviewer #1: Yes

Reviewer #3: No

6. Review Comments to the Author

Reviewer #1: The grammar needs a bit improvement. The S. aureus should be uniformly italic across the article. The rest, I have no more comments.

Reviewer #3: The manuscript addresses an important public health issue by investigating Methicillin-resistant Staphylococcus aureus along the beef production chain. The integration of phenotypic resistance profiling with mecA phylogenetic analysis is valuable and relevant. However, the manuscript requires careful revision to improve clarity, consistency, and scientific accuracy. Several sections contain repetition, inconsistent formatting (particularly for scientific names, gene names, and antibiotic nomenclature), and methodological ambiguities. Addressing these issues will significantly enhance the overall quality and readability of the manuscript.

Lines 166–167: Reference 32 (Mehrotra et al., 2000) does not correspond to the same primers authors used for mecA amplification. The primers described in the manuscript differ from those in the cited reference. Please provide an appropriate and accurate reference.

Additionally, Please provide a clearer, high-resolution PCR gel image and verify the reported product size.

Lines 104–109: This section repeats information presented in the following paragraph. Please remove the redundant text to improve clarity and avoid duplication.

Line 136: The term “Gram” should be written in lowercase (i.e., “gram”).

Lines 144–146: Please group antibiotics according to their respective classes and clearly mention the class names. Additionally, ensure consistency in formatting—antibiotic names should follow a uniform style throughout the manuscript (e.g., consistent capitalization).

Line 152: Scientific names should be written in italics.

Line 160: The sentence is unclear. It is recommended to revise it as: “Genomic DNA was extracted from Staphylococcus aureus.”

Line 183: The DNA extraction method has already been described earlier. Repetition is unnecessary—please remove this section.

Line 250: Scientific names should be written in italics.

Line 251: The gene name mecA is inconsistently formatted (italicized in some places and not in others). Please adopt a consistent style throughout the manuscript.

7. PLOS authors have the option to publish the peer review history of their article (what does this mean?). If published, this will include your full peer review and any attached files.

Reviewer #1: **Yes:** Tahir Usman

Reviewer #3: No

---

## [Author Response · Author response to Decision Letter 2]

17 Apr 2026

Dear Editor-in-Chief and Reviewers,

We are thankful for the consideration of our manuscript in your journal and for the constructive comments provided, which have significantly improved the quality and clarity of the manuscript.

---

## [Decision Letter · Decision Letter 2]

20 Apr 2026

Methicillin-Resistant Staphylococcus aureus Along the Beef Production Line: Phenotypic Resistance and mecA Phylogeny in Two Ethiopian Municipal Abattoirs

PONE-D-25-53029R2

Dear Dr. Abey,

We’re pleased to inform you that your manuscript has been judged scientifically suitable for publication and will be formally accepted for publication once it meets all outstanding technical requirements.

Kind regards,

Md. Tanvir Rahman, DVM, MSc (Canada), PhD (UK), FBAS

Academic Editor

PLOS One

Additional Editor Comments (optional):

Thanks for the revision.

Reviewers' comments:

Reviewer's Responses to Questions

**Comments to the Author**

1. If the authors have adequately addressed your comments raised in a previous round of review and you feel that this manuscript is now acceptable for publication, you may indicate that here to bypass the “Comments to the Author” section, enter your conflict of interest statement in the “Confidential to Editor” section, and submit your "Accept" recommendation.

Reviewer #3: All comments have been addressed

2. Is the manuscript technically sound, and do the data support the conclusions?

Reviewer #3: Yes

3. Has the statistical analysis been performed appropriately and rigorously? 

Reviewer #3: I Don't Know

4. Have the authors made all data underlying the findings in their manuscript fully available?

Reviewer #3: Yes

5. Is the manuscript presented in an intelligible fashion and written in standard English?

Reviewer #3: Yes

6. Review Comments to the Author

Reviewer #3: The authors have adequately addressed all comments and made the required revisions. The manuscript has improved and is now suitable for publication.

7. PLOS authors have the option to publish the peer review history of their article (what does this mean?). If published, this will include your full peer review and any attached files.

Reviewer #3: No

---

## [Editor Report · Acceptance letter]

PONE-D-25-53029R2

PLOS One

Dear Dr. Abey,

I'm pleased to inform you that your manuscript has been deemed suitable for publication in PLOS One. Congratulations! Your manuscript is now being handed over to our production team.

Kind regards,

on behalf of

Professor Md. Tanvir Rahman

Academic Editor

PLOS One